# Setting up a Virtual Calprotectin Clinic in Inflammatory Bowel Diseases: Literature Review and Nancy Experience

**DOI:** 10.3390/jcm9092697

**Published:** 2020-08-20

**Authors:** Ferdinando D’Amico, Patrick Netter, Cedric Baumann, Muriel Veltin, Camille Zallot, Isabelle Aimone-Gastin, Silvio Danese, Laurent Peyrin-Biroulet

**Affiliations:** 1Department of Biomedical Sciences, Humanitas University, Pieve Emanuele, 20090 Milan, Italy; damico_ferdinando@libero.it (F.D.); sdanese@hotmail.com (S.D.); 2Department of Gastroenterology and Inserm NGERE U1256, University Hospital of Nancy, University of Lorraine, 54500 Vandoeuvre-lès-Nancy, France; muriel.veltin@micilor.com (M.V.); c.zallot@chru-nancy.fr (C.Z.); 3Ingénierie Moléculaire et Ingénierie Articulaire (IMoPA), UMR-7365 CNRS, Faculté de Médecine, University of Lorraine and University Hospital of Nancy, 54000 Nancy, France; gotheilnetter@yahoo.fr; 4Clinical Research Support Facility, Methodological and Statistical Unit, University Hospital of Nancy, 54000 Nancy, France; c.baumann@chru-nancy.fr; 5Department of Biochemistry-Molecular Biology-Nutrition, Nancy University Hospital, 54000 Nancy, France; i.gastin@chru-nancy.fr; 6Inserm UMR_S1256 N-GERE, Nutrition-Genetics-Environmental Risk Exposure, University Hospital of Nancy, University of Lorraine, 54500 Vandoeuvre-lès-Nancy, France; 7IBD center, Department of Gastroenterology, Humanitas Clinical and Research Center—IRCCS, Rozzano, 20089 Milan, Italy

**Keywords:** virtual clinic, fecal calprotectin home test, IBDoc^®^, inflammatory bowel disease

## Abstract

Technological progress, including virtual clinics, web or smartphone-based applications, and assessment of fecal calprotectin (FC) at home has favored the implementation of treat to target strategies for patients with inflammatory bowel diseases (IBD). Although these innovations are promising and have been associated with a significant reduction in health costs, their application in clinical practice is limited. Here, we summarize the most recent literature on virtual clinics and available FC home tests. In addition, we report the experience of IBD patients monitored through the IBDoc^®^ test at the Nancy University Hospital, focusing on usability testing and patient’s satisfaction. This pilot experience shows that a virtual calprotectin clinic doubles adherence rate to FC in IBD patients. This finding is especially clinically relevant in the post-coronavirus disease 2019 (COVID-19) pandemic era, with an increasing use of e-health.

## 1. Introduction

Crohn’s disease (CD) and ulcerative colitis (UC) are inflammatory bowel diseases (IBD) with a remitting and relapsing course [1,2]. The prevalence of IBD exceeds 0.3% in the general population of North America and Europe and is increasing considerably in the newly industrialized countries [3]. IBD are chronic conditions requiring periodic follow-up to monitor disease activity and to prevent an uncontrolled inflammation that could lead to negative disease outcomes such as recurrence, hospitalization, and surgery [1,4,5]. The management of IBD patients is based on outpatient visits, biochemical tests (e.g., blood and fecal tests), endoscopic procedures, and imaging techniques (e.g., small bowel ultrasound, computed tomography, and magnetic resonance imaging), and there is accumulating evidence that tight monitoring of patients is the best strategy for optimal care [6]. However, this strategy determines a great economic burden on the health system and leads to a progressive increase in outpatient visits and procedures, causing delays in planning health services and lengthening waiting lists [7,8]. In addition, given the unpredictable nature of IBD, patients often experience relapses occurring far from their next scheduled follow-up and are forced to perform urgent visits [9]. This patient management is associated with a significant loss of productivity not only for patients but also for their caregivers, underlining the need for new solutions and new approaches [10]. In recent years, technological development has provided a promising solution for improving patient care through the introduction of information and communication technologies (ICTs), known as “e-health” [11]. E-health is a new strategy including several components such as the use of web-based or smartphone/social media-based applications and virtual clinics, which allow physicians to provide assistance to patients remotely [11]. E-health has been associated with lower rates of hospitalizations and medical visits and could have a substantial impact on healthcare costs [12]. Moreover, it has also been applied to some procedures such as the measurement of fecal calprotectin (FC) allowing the home dosage of this biomarker, which is frequently used in daily practice [13]. The FC home measurement does not involve the transport of feces to the laboratory and could improve patient adherence to the test compared to traditional method, which is approximately 40% [14,15]. To date, despite the conceivable benefits associated with the widespread adoption of these innovations, their use in clinical practice is limited and some doubts about their usefulness remain [16]. The aims of our work are to summarize the literature on e-health and FC home tests in the IBD field and report the experience of the Nancy University Hospital with the e-health strategy in IBD patients.

### 1.1. Efficacy of Patient Care Based on the e-Health

The different aspects of the e-health strategy have been evaluated in both CD and UC patients. Elkjaer and colleagues conducted a randomized trial [17] in Denmark and Ireland to assess the feasibility of telemedicine in patients with UC. In total, 233 patients were randomized to traditional care or a web-based approach, which consisted of contacting the physician if predetermined symptoms occurred. Of note, most patients in the telemedicine group (88%) preferred the new approach over the traditional one and a significant increase in adherence to UC treatment was found in the experimental group compared to the control group (31% in Denmark and 44% in Ireland). In addition, the number of routine outpatient clinic and additional visits due to UC relapses was significantly lower in the telemedicine group compared to controls during the 12-month study period (35 vs. 92 and 21 vs. 107, respectively, *p* < 0.0001), resulting in an overall cost saving of 189 €/patient/year. A randomized controlled trial by Del Hoyo et al. [18] evaluated the impact of virtual clinic on outcomes of 63 IBD patients. Clinical remission rate of patients managed at home through the telemonitoring of Crohn’s disease and ulcerative colitis (TECCU) web-system at Week 24 was compared with that of subjects undergoing a traditional face-to-face visit or a nurse-assisted telephone care. Patients were randomly assigned 1:1:1 into three study arms. After 24 weeks, the TECCU group achieved a higher rate of remission compared to the other groups (81% vs. 71.4% and 66.7%, respectively). No difference in terms of medication-related adverse events, hospitalization, and surgery was found among the study arms, suggesting that home monitoring was safe and could be a viable strategy for patient management. A Dutch study [19] investigated the effects of a telemedicine system (myIBDcoach) on disease outcomes of 909 IBD patients (544 CD and 365 UC). After 12 months, no difference between telemedicine group and standard care was found in the rate of flares, use of steroids, emergency visits, and IBD-related surgeries. However, significantly fewer outpatient visits per patient (1.26 vs. 1.98, *p* < 0.0001) and hospitalizations (16 vs. 29, *p* = 0.046) were recorded in patients using myIBDcoach. Moreover, in a pos thoc analysis of this study [20], a significant reduction in both direct and indirect costs was revealed in IBD patients managed by telemedicine compared to the traditional group (mean annual reduction of € 547/patient, 95% CI from €−1029 to €2143). Another randomized trial [12] investigated disease activity and quality of life of 348 IBD patients (236 CD and 112 UC) receiving telemedicine or standard care during a 12-month period. Interestingly, the rate of IBD-related hospitalizations increased in the standard group (from 14.7 to 16.4), while a significant reduction was detected in patients receiving telemedicine weekly (from 24.1 to 9.8, *p* = 0.04). Improvements in quality of life and disease activity were recorded in both study populations, but no significant difference was identified between the groups.

### 1.2. A System for Monitoring Fecal Calprotectin at Home

FC home tests allow IBD patients to independently measure FC levels from stool samples at home. Each FC assay kit includes a paper stool catcher to be placed on the edges of the toilet bowl, a specific extraction instrument, and a manufacturer-specific test cassette [21,22,23]. Patients must have smartphones compatible with the measurement test and download a specific test application to perform data analysis [21,22,23]. After collecting the feces, patients must use the extraction tool to collect a predefined amount of stool and extract the calprotectin (Figure 1) [21,22,23]. Subsequently, the extraction tool is applied on the reading cassette and the content is released [21,22,23]. At this point, the smartphone camera is positioned in front of the cassette and is used to read the results [21,22,23]. FC value is displayed on the smartphone and result interpretation is reported according to manufacturer’s guidelines (e.g., normal, moderate, or high) [21,22,23]. At the end, the result is forwarded to the patient’s healthcare team who can remotely monitor the FC levels and make diagnostic-therapeutic decisions [21,22,23]. It is important to note that FC home analysis provides a semi-quantitative measurement, varying according to the specific test used: from <30.0 to >1000.0 μg/g for IBDoc^®^ (Bühlmann Laboratories AG, Schönenbuch, Switzerland) [24], from 25 to 2000 μg/g for QuantonCal^®^ (Preventis GmbH, Bensheim, Germany) [25], and from 70 to 1500 μg/g for CalproSmart^TM^ (CALPRO AS, Lysaker, Norway) [23]. Finally, based on the result of FC measurement, the physician decides whether to continue the patient’s follow-up unchanged, modify the treatment, or perform further investigations (e.g., endoscopic or radiological procedures).

### 1.3. Fecal Calprotectin Home Testing

Recently, increasing attention has been paid to FC and several tools have been developed to measure FC directly at home and to improve test compliance (e.g., IBDoc^®^, QuantonCal^®^, and CalproSmart^TM^) [26]. The feasibility of the home FC measurement was assessed in a randomized controlled trial enrolling 123 IBD patients (31 CD, 89 UC, and 3 unclassified IBD) [15]. Experimental group patients were monitored at home with CalDetect tool, while a traditional enzyme-linked immunosorbent assay (ELISA) test was adopted in the control group [15]. Compliance, defined as participation in the study for more than six months, was greater in the home monitoring group than controls (71% vs. 46%, *p* = 0.005) [15]. Although compliant patients were older (mean age: 38 years vs. 31 years, 95% CI: 35–40 and 26–36, *p* = 0.013) and more symptomatic than non-compliant patients, no factor predicted adherence to treatment [15]. The number of relapses and hospitalizations (25 vs. 21 and 4 vs. 1, respectively, *p* > 0.05 for both comparisons), and patients’ quality of life did not differ between the two groups, but the number of gastroenterological visits was significantly lower in subjects managed at home compared to the traditional group during a study period of 12 months (24% vs. 40% *p* = 0.05) [15]. A randomized trial evaluated the non-inferiority of IBDoc^®^ compared to standard care in 73 CD patients [27]. No difference in terms of symptoms and quality of life was reported between the two groups after 12 months [27]. Importantly, FC dosing adherence was not optimal as only 30% of patients completed all scheduled measurements [27]. Despite this, over 50% of IBDoc^®^ users still wanted to use it in the future and suggested its use to other people [27]. Another study by Bello et al. enrolled 58 IBD patients (40 CD and 18 UC) showing an IBDoc^®^ adherence rate of approximately 47% over an eight-week follow-up period (five measurements were expected) [28]. A good correlation was found between IBDoc^®^ and ELISA test (intra-class correlation coefficient = 0.88) and the home test had high diagnostic accuracy in predicting a FC > 300 μg/g with the ELISA test (sensitivity: 89.8%, specificity: 95.5%, negative predictive value: 91.4%, and positive predictive value: 94.6%) [28]. Reproducibility of the extraction and reading procedures were acceptable with a mean individual coefficient of variation of 17.5% (range: 0–42%). In addition, home measurement had high usability for patients, defined as the ability to understand the method and use the extraction tool, reading cassette, and smartphone camera [28]. A study by Wei et al. enrolling 51 IBD patients (23 CD, 27 UC, and 1 unclassified IBD) showed an 80% correlation between IBDoc^®^ and Quantum Blue when a 250 μg/g cut-off was used [29]. Of note, more than 90% of patients believed that the home test was simple to use and were satisfied after the measurement, reporting that they preferred IBDoc^®^ over the traditional test (86% vs. 14%) [29]. On the other hand, different IBDoc^®^ coefficients of variation were reported depending on FC threshold, ranging from 4.8% for FC < 50 μg/g to 26.6% if FC was greater than 200 μg/g [30]. Variability and accuracy of another home test, the CalproSmart^TM^, were evaluated in a randomized clinical trial comparing the home test with the ELISA test [31]. In total, 221 IBD patients were recruited (115 UC and 106 CD). CalproSmart^TM^ showed adequate intra- and inter-assay coefficients of variation (4.42% and 12.49%, respectively), valid intra- and inter-assay reproducibility (4.6% and of 7.1%, respectively), and high diagnostic accuracy in predicting disease activity (sensitivity of 82% and specificity of 85% with a FC cut-off of 150 μg/g) [31]. Additionally, a randomized trial [32] by Ankersen et al. compared the outcomes of 102 IBD patients (23 CD, 74 UC, and 5 unclassified IBD) monitored through CalproSmart^TM^ every 3 months or on demand during one-year follow-up. No difference was found between the two arms in terms of relapse time, remission time, disease course, compliance with medical therapy, quality of life, and patient satisfaction, suggesting that both strategies were valid and on demand approach could be preferred in a context of healthcare cost saving [32].

### 1.4. The Nancy Experience

The Nancy University Hospital (Vandoeuvre-lès-Nancy, France) is the home of the Lorraine IBD Network (Le réseau lorrain des Maladies Inflammatoires Chroniques Intestinales, MICILOR), which follows over 2500 IBD patients. Starting from 2016, the e-health system was associated with the usual face-to-face activity, including virtual clinics and a dedicated hotline, active 24/7. A specialized IBD nurse (MV) manages the hotline by receiving on average 5 calls and 10 emails per day. The most frequent reasons for using the hotline are to request information and to communicate the worsening of symptoms or the results of diagnostic tests. The nurse has the role of filtering the information to the physicians, who decide on any changes in patient management (e.g., to anticipate a visit, perform a virtual clinic, or request hospitalization). In addition, in some cases, after an initial contact mediated by the IBD nurse, direct communication between patient and physician is also possible. Finally, starting from 2018, the possibility of performing the FC test at home was introduced.

## 2. Methods

We designed a pilot prospective observational cohort study to evaluate adherence to measurement, patient satisfaction, and usability after FC home monitoring through the validated IBDoc^®^ test (Bühlmann, Switzerland) [33]. We recruited 30 consecutive adult patients with confirmed CD or UC based on standardized clinical, endoscopic, histological, and radiological criteria, who were referred for a standard outpatient IBD care at the Nancy University Hospital (France) between September 2018 and December 2019. Enrollments were performed only one day a week. All patients with a compatible smartphone who agreed to monitor FC at home through IBDoc^®^ were eligible for inclusion. Patients who already had a prescription for FC measurement with the traditional method were excluded. A specialized IBD nurse (MV) provided patients with all information about IBDoc^®^ use at enrollment All patients were asked to perform a FC dosage at home through IBDoc^®^ and to fill out a questionnaire to evaluate IBDoc^®^ usability and their satisfaction with the test. Patient demographics and disease characteristics (sex, age, residence, marital status, educational level, IBD type, and disease duration) were also collected. The questionnaire was initially developed in French and was later translated into English by native English speakers. The questionnaire was based on multiple-choice questions and participants were asked to numerically evaluate usability and their satisfaction with the test through a Likert scale from 1 to 7 (where 1 indicated the most negative value and 7 the most positive value). Values between 1 and 3 indicated a negative result, those between 5 and 7 a positive result, and values equal to 4 represented an intermediate value. Importantly, FC was considered normal for values < 100 μg/g, while it was classified as moderate or high for thresholds >100 or >300 μg/g, respectively, according to manufacturer’s indications [28,34]. The ethical approval code of our study was reported to the Commission Nationale de l’Informatique et des Libertés (number 1,404,720).

### Statistical Analysis

Categorical variables were described by counts and percentages, while continuous variables by mean ± standard deviation. Univariate and multivariate regressions were performed to explore factors influencing patients’ satisfaction with IBDoc^®^ use. All patient demographics and disease characteristics (sex, age, residence, marital status, educational level, IBD type, and disease duration) were tested in a univariate conditional regression model to identify the profile of satisfied patients after IBDoc^®^ use. Variables with a *p*-value < 0.1 in univariate analysis were candidate for a multivariate conditional regression. Associations were described by matched odd ratios (mORs) with 95% confidence interval (CI). The threshold for statistical significance was fixed at 5% for each test. All data processing and statistical analyses were performed using SAS software (version 9.4, SAS Institute Inc., Cary, NC, USA).

## 3. Results

All patients agreed to participate but only 20 (66.7%) performed the test and were included in our study. The causes of non-adherence were: difficulty in performing the measurement alone (3/10, 30%), absence of an economic reimbursement (2/10, 20%), disease worsening (2/10, 20%), forgetfulness (2/10, 20%), and address change (1/10, 10%). Characteristics of included patients are shown in Table 1. Most patients were women (12/20, 60%) and over three quarters of patients were aged between 25 and 44 years (16/20, 80%). The remaining patients were less than 25 years old (4/20, 20%). Most patients were unmarried (15/20, 75%) and had a high level of education (>Bachelor’s degree) (14/20, 70%). About half of the patients lived in rural areas (11/20, 55%), while the other subjects lived in urban areas (9/20, 45%). Thirteen patients (65%) suffered from CD and 7 (35%) from UC. Several ranges of disease duration were reported: two patients had IBD for less than 1 year (10%), seven for 1–5 years (35%), five for 6–10 years (25%), and six for more than 11 years (30%). An unsatisfactory (3/20, 15%) or quite unsatisfactory (4/20, 20%) quality of life was found in about a third of the participants. A quarter of individuals were members of an association for IBD patients (5/20, 25%). All patients reported regular use of the smartphone (20/20, 100%).

### 3.1. Information about Fecal Calprotectin and IBDoc^®^ Test

Stool sampling was considered difficult in only two cases (10%). Half of the patients (10/20, 50%) knew what FC was and most of them had a prescription for FC test (8/10, 80%) (Table 2). All patients were informed of the existence of the FC home test by their gastroenterologist at the time of inclusion visit (20/20, 100%) and common opinion was that FC test was used to avoid colonoscopy (9/20, 45%), detect early disease relapse (9/20, 45%), or improve therapy adaptation (6/20, 30%).

### 3.2. Information about IBDoc^®^ Usability

All participants reported that they never used IBDoc^®^ before (20/20, 100%) and received adequate information regarding its use (20/20, 100%). Installation and connection of the application were mostly considered easy (19/20, 95%). Similarly, test preparation was frequently classified as easy (18/20, 90%). On the other hand, stool sampling was rated as difficult in a quarter of respondents (5/20, 25%), and one fifth of patients defined stool preparation as difficult (4/20, 20%). Only one patient (5%) reported that it was difficult to deposit the stool in the cassette, while no difficulty was found in cassette reading, recording, and transmission of the results. Time analysis was recognized as appropriate by most individuals (19/20, 95%). In more than two-thirds of patients (14/20, 70%), IBDoc^®^ results were associated with subjective clinical status, while, in a few cases, biochemical values and symptoms were unpaired (5/20, 25%).

### 3.3. Information about Patients’ Satisfaction with IBDoc^®^

Most patients were satisfied with IBDoc^®^ (17/20, 85%). No patient was dissatisfied. Overall, most patients rated the use of the home test as easy (19/20, 95%). In no case was IBDoc^®^ use considered difficult. Finally, all patients would have recommended other people to adopt IBDoc^®^ (20/20, 100%) and all but one (19/20, 95%) reported that they wanted to regularly use FC home test in the future.

### 3.4. Fecal Calprotectin Values and Profile of Patients Satisfied with IBDoc^®^

Mean FC value was 366.1 μg/g (±standard deviation (SD) = 374.2). Most patients had high FC values (8/20, 40%), while moderate (5/20, 25%) or normal (7/20, 35%) levels were found in a lower percentage of cases. No patient demographics or disease characteristics were significantly associated with patients’ satisfaction in the univariate analysis preventing the definition of a profile of patients satisfied with IBDoc^®^.

## 4. Discussion

FC measurement has become increasingly important for the management of IBD patients and it is consistently used in daily clinical practice. FC test is recommended by ECCO guidelines to assess disease activity, evaluate response to therapies, and predict the early onset of relapses [35]. Interestingly, a French survey including 916 IBD patients reported that fecal tests were more accepted than colonoscopy (*p* < 0.0001), but adherence to fecal test was considerably limited by stool sample transportation to the laboratory [36]. FC home test does not require the transport of the sample to the laboratory, effectively eliminating this important limitation towards patient compliance. In our experience, we investigated patients’ views regarding usability and satisfaction with IBDoc^®^. Two thirds of enrolled patients performed the test resulting in double the adherence compared to a previous study of our group showing an adherence rate of 35% with the ELISA test [14]. The test was classified as simple to use by most users (95%) and the percentage of satisfied patients was very high (95%). In addition, all patients reported that they wanted to recommend the test to other patients and regularly use the FC home dosage. Only one patient did not prefer to continue using the test as IBDoc^®^ was considered too expensive. However, it is important to underline that the actual price of IBDoc^®^ is comparable to ELISA test (€30 vs. €41 approximately) [37]. Of note, we tried to define the profile of patients satisfied with IBDoc^®^ use to identify subjects to be monitored with this test. Unfortunately, no predictor of satisfaction was found. This could be explained by a lack of power of the analyses, not allowing to exclude that some tested factors could be really linked to satisfaction. To date, it is not known how to select patients to be monitored with FC home tests or traditional tests. A study including 101 pediatric and adult IBD patients compared the diagnostic accuracy of IBDoc^®^, Quantum Blue, and ELISA tests [37]. A good correlation was reported among the three methods although it was greater when low FC concentrations (<500 μg/g) were examined compared to higher, concentrations (91% and 71% vs. 81% and 64%, respectively) [37]. Similarly, there are insufficient data to establish which is the best FC home test. A head-to-head trial compared the diagnostic accuracy of three home tests (IBDoc^®^, QuantOnCal^®^, and CalproSmart^TM^) and three ELISA tests [26]. A high percentage of agreement with all home tests (87%, 82%, and 76%) was reached when low FC levels (<500 μg/g) were considered, while a significantly lower concordance (37%, 19%, and 37% respectively) was found with high FC values (>500 μg/g) [26]. Fewer read errors were detected through IBDoc^®^ than CalproSmart^®^ and QuantOnCal^®^ (1.9% vs. 5.8% and 4.8%, *p* = 0.002 and *p* = 0.012, respectively), but no tool was superior to the others [26]. These studies showed a high variability between the available tools indicating that they were not interchangeable. For this reason, FC monitoring should always be performed with the same method to avoid measurement variability [13]. In the absence of clear evidence demonstrating the superiority of one test over the others, it is legitimate to argue that patient’s preference plays an important role in the decision. Thus, our study is clinically relevant as it provides information on patients’ opinions, suggesting that IBDoc^®^ may be a valid option for patients’ follow-up. Moreover, the use of FC home tests is of great relevance in the context of the current health emergency caused by the coronavirus disease 2019 (COVID-19) that has caused thousands of deaths worldwide [38]. In fact, due to the total lockdown and to the likely fecal–oral transmission of the infection, a reduction in or total suspension of the request for FC dosage has been reported by about 50% of physicians [39]. E-health, including both virtual clinic and virtual calprotectin, have been extremely important during the pandemic, proving to be a valid alternative to the usual face-to-face hospital management [39]. At the time of writing, most countries have overcome the lockdown and are gradually resuming their pre-COVID-19 activities. However, it should be noted that, in the post-pandemic period, precautions will still be needed to prevent new infections. In addition, the number of patients to be monitored will be particularly high, as many visits or diagnostic tests have been canceled or postponed in the previous months. Therefore, it would be desirable to stratify patients according to disease activity and the FC home test could be a valuable tool for monitoring patients in clinical remission [40]. Additional studies comparing home tests and traditional ELISA test as well as the correlation with endoscopic disease activity are needed before these tools are widely used in clinical practice. Some limitations of our study need to be addressed. Firstly, we included a small number of patients. Our study population could be sufficient to provide indicative information about satisfaction and usability of the IBDoc^®^, but it did not allow identifying factors associated with patient satisfaction. Secondly, all patients were younger than 50 years and had an average high degree of education, suggesting that an interpretation bias for usability cannot be excluded. Thirdly, although the Likert measurement scale is widely adopted for the evaluation of real-life studies, the non-use of validated tools such as the system usability scale represents a weakness of our work [41]. Fourthly, the involvement of a dedicated specialized nurse, which ensured adequate training for all patients, could have influenced the positive responses of the users. Reproducibility of our data should be confirmed in other centers without dedicated personnel. Furthermore, large studies are needed to support high usability, satisfaction, and patient adherence with IBDoc^®^, and to compare IBDoc^®^ results with other FC home tests in order to identify the best tool to be used in IBD patients.

## 5. Conclusions

The IBDoc^®^ is a simple tool to use and high satisfaction is found among IBDoc^®^ users. IBD patients should be adequately informed and trained on the use of this test. FC home tests are an additional value for e-health approach in IBD patients. In the near future, these tests could allow not only tight monitoring of IBD patients but also their greater involvement in disease management.

## Figures and Tables

**Figure 1 jcm-09-02697-f001:**
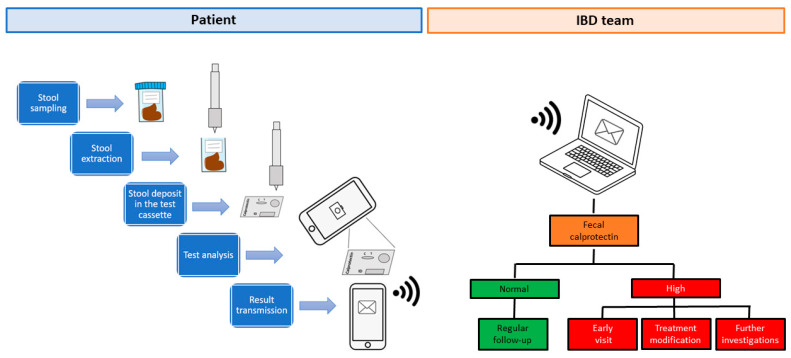
Multi-step procedure for measuring fecal calprotectin at home with IBDoc^®^ test. IBD: inflammatory bowel diseases.

**Table 1 jcm-09-02697-t001:** Patients’ characteristics.

Patients’ Characteristics	n (%)
Patients	20
Female	12 (60%)
Age	
<25 years	4 (20%)
25–44 years	16 (80%)
Marital status:	
Married	5 (25%)
Unmarried	15 (75%)
Residence	
Urban area	9 (45%)
Rural area	11 (55%)
Educational level	
<Bachelor’s degree	1 (5%)
Bachelor’s degree	5 (25%)
>Bachelor’s degree	14 (70%)
Disease	
Ulcerative colitis	7 (35%)
Crohn’s disease	13 (65%)
Disease duration	
<1 year	2 (10%)
1–5 years	7 (35%)
6–10 years	5 (25%)
>11 years	6 (30%)
CD location	
Ileitis	3 (23%)
Colitis	1 (8%)
Ileocolitis	9 (69%)
UC location	
Extensive colitis	5 (71%)
Left-side colitis	2 (29%)
Proctitis	0
Smoking status	
Active smoker	5 (25%)
Former smoker	1 (5%)
Non-smoker	14 (70%)
Perianal disease	3 (15%)
Upper disease	1 (5%)
Surgery	6 (30%)
Clinical disease activity	
Harvey-Bradshaw Index, mean ± standard deviation	2.15 ± 1.72
Partial Mayo score, mean ± standard deviation	2.42 ± 2.63
Medications	
Local steroid	6 (30%)
Systemic steroid	12 (60%)
5-ASA	8 (40%)
Thiopurine	9 (45%)
Methotrexate	4 (20%)
Infliximab	9 (45%)
Adalimumab	11 (55%)
Vedolizumab	4 (20%)
Ustekinumab	4 (20%)
Current quality of life	
7 Completely satisfactory	4 (20%)
6 Satisfactory	3 (15%)
5 Quite satisfactory	3 (15%)
4 Indifferent	3 (15%)
3 Quite unsatisfactory	4 (20%)
2 Not satisfactory	3 (15%)
1 Not at all satisfactory	0
Membership in an association for IBD patients	
Yes	5 (25%)
No	15 (75%)
Frequency of smartphone use	
7 Completely regular	11 (55%)
6 Regular	3 (15%)
5 Quite regular	6 (30%)
4 Indifferent	0
3 Quite not regular	0
2 Not regular	0
1 Not at all regular	0

5-ASA: mesalazine.

**Table 2 jcm-09-02697-t002:** Information about fecal calprotectin test, usability, and patients’ satisfaction with IBDoc^®^ use.

Questionnaire	n (%)
Information about Fecal Calprotectin and IBDoc^®^	
(1) Is stool sampling a problem for you?	
7 Definitely simple	12 (60%)
6 Simple	5 (25%)
5 Quite simple	0
4 Indifferent	1 (5%)
3 Quite difficult	2 (10%)
2 Difficult	0
1 Definitely difficult	0
(2) Have you heard of the fecal calprotectin dosage?	
Yes	10 (50%)
No	10 (50%)
(3) Did you already have a prescription for fecal calprotectin dosage with traditional method?	
Yes	8/10 (80%)
No	2/10 (20%)
(4) Who told you about the IBDoc^®^ test?	
Gastroenterologist	20 (100%)
General practitioner	0
Pharmacist	0
Participation in information days	0
Newspapers/magazines	0
Internet	0
(5) What is the IBDoc^®^ test for (multiple answers are allowed)?	
To avoid a colonoscopy	9 (45%)
To detect early disease relapse	9 (45%)
To improve therapy adaption	6 (30%)
To manage fecal calprotectin test independently	3 (15%)
To monitor disease activity	1 (5%)
Information about IBDoc^®^ usability	
(6) Is it the first time you use IBDoc^®^?	
Yes	20 (100%)
No	0
(7) Do you think you have received adequate information for the use of IBDoc^®^?	
7 Definitely adequate	18 (90%)
6 Adequate	2 (10%)
5 Quite adequate	0
4 Indifferent	0
3 Quite inadequate	0
2 Inadequate	0
1 Definitely inadequate	0
(8) How do you rate the application installation and connection?	
7 Definitely easy	8 (40%)
6 Easy	9 (45%)
5 Quite easy	2 (10%)
4 Indifferent	0
3 Quite difficult	1 (5%)
2 Difficult	0
1 Definitely difficult	0
(9) How do you rate test preparation?	
7 Definitely easy	5 (25%)
6 Easy	10 (50%)
5 Quite easy	0
4 Indifferent	3 (15%)
3 Quite difficult	1 (5%)
2 Difficult	0
1 Definitely difficult	1 (5%)
(10) How do you rate stool sampling?	
7 Definitely easy	6 (30%)
6 Easy	7 (35%)
5 Quite easy	2 (10%)
4 Indifferent	0
3 Quite difficult	2 (10%)
2 Difficult	3 (15%)
1 Definitely difficult	0
(11) How do you rate stool preparation?	
7 Definitely easy	8 (40%)
6 Easy	8 (40%)
5 Quite easy	0
4 Indifferent	0
3 Quite difficult	3 (15%)
2 Difficult	1 (5%)
1 Definitely difficult	0
(12) How do you rate the stool deposit in the test cassette?	
7 Definitely easy	11 (55%)
6 Easy	7 (35%)
5 Quite easy	1 (5%)
4 Indifferent	0
3 Quite difficult	1 (5%)
2 Difficult	0
1 Definitely difficult	0
(13) How do you rate reading of the test cassette?	
7 Definitely easy	10 (50%)
6 Easy	7 (35%)
5 Quite easy	2 (10%)
4 Indifferent	1 (5%)
3 Quite difficult	0
2 Difficult	0
1 Definitely difficult	0
(14) How do you rate recording of test results?	
7 Definitely easy	16 (80%)
6 Easy	4 (20%)
5 Quite easy	0
4 Indifferent	0
3 Quite difficult	0
2 Difficult	0
1 Definitely difficult	0
(15) How do you rate time analysis of stools?	
7 Definitely easy	13 (65%)
6 Easy	6 (30%)
5 Quite easy	0
4 Indifferent	1 (5%)
3 Quite difficult	0
2 Difficult	0
1 Definitely inappropriate	0
(16) How do you rate results’ transmission?	
7 Definitely easy	16 (80%)
6 Easy	4 (20%)
5 Quite easy	0
4 Indifferent	0
3 Quite difficult	0
2 Difficult	0
1 Definitely difficult	0
(17) Do the IBDoc^®^ results match the clinical status?	
7 Definitely easy	11 (55%)
6 Easy	1 (5%)
5 Quite easy	2 (10%)
4 Indifferent	1 (5%)
3 Quite difficult	4 (20%)
2 Difficult	1 (5%)
1 Definitely unpaired	0
Information about patients’ satisfaction with IBDoc^®^	
(18) How satisfied are you with the use of IBDoc^®^?	
7 Definitely easy	11 (55%)
6 Easy	6 (30%)
5 Quite easy	2 (10%)
4 Indifferent	1 (5%)
3 Quite difficult	0
2 Difficult	0
1 Completely unsatisfied	0
(19) Overall, is the IBDoc^®^ test easy to use?	
7 Definitely easy	7 (35%)
6 Easy	8 (40%)
5 Quite easy	4 (20%)
4 Indifferent	1 (5%)
3 Quite difficult	0
2 Difficult	0
1 Definitely difficult	0
(20) Would you like to use the IBDoc^®^ test regularly?	
Yes	19 (95%)
No	1 (5%)
(21) Would you recommend the use of IBDoc^®^ test to other patients?	
Yes	20 (100%)
No	0

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
