# Peer review of "Setting up a Virtual Calprotectin Clinic in Inflammatory Bowel Diseases: Literature Review and Nancy Experience"

_jcm, 2020, doi:10.3390/jcm9092697_

Round 1

Reviewer 1 Report

Brief Summary: This manuscript by D’Amico et al. provides a narrative review of e-health and fecal calprotectin home testing kits. In addition, they provide their own survey study results relating to the use of home fecal calprotectin kits. This review is timely due to the impact that COVID-19 has had on healthcare and the management of IBD, as more care is shifting remotely.

General Comments: The strengths of this manuscript relate to its combination of a nice review of the literature, specifically as it relates to home fecal calprotectin testing, with the additional experience of a large IBD center in this area of study. The primary limitation however is the small sample size of the survey which limits its generalizability and underpowers statistical analysis of the data. Furthermore, some improvement in the grammar and sentence structure is needed. Consideration may be given to try to increase the number of individuals who answer the survey either within your institution or by incorporating other institutions as this would strengthen the survey results.

Specific comments:

Line 34- "adhesion" should be changed to "adherence"

Line 73- "E-Health Literature Evidence" should be rephrased.

Line 81- Are the "routine" outpatient visits that are referred to referring to all outpatient visits inclusive of telemedicine visits, or is this referring to additional visits? It would be helpful to clarify this point.

Line 106- "Fecal Calprotectin Home Test System" should be rephrased.

Line 125- "Fecal Calprotectin Home Test Literature Evidence" should be rephrased.

Line 170- It is interesting that in following over 2500 patients a year there are only an average of 5 calls and 10 emails per day. This seems low. Is this in addition to communication between patients and their doctors directly. If so, would add clarification about this.

Line 195- Could more information be given on the fecal calprotectin thresholds chosen. These thresholds don’t seem to align with growing evidence for cutoffs in CD and UC.

Line 209- It is stated that 30 subjects agreed to participate. How many people were asked to participate? Should include the response rate.

Line 220- Smart phone use was broken down into completely regular, regular or quite regular use. It is unclear what the difference between these groups are. Clarification of what these designations mean would be helpful.

Line 223- It is unclear what difficulty in handling the stool means. Clarification in the manuscript regarding this would be helpful.

Line 231- It is unclear what the difference between definitely adequate or adequate information is. Was this defined in the survey for the patients. Would recommend clarifying this in the manuscript. Similar statements are made relating to ease of use and difficulty with stool sampling as well as other survey results. Clarification of this grading scale is needed as it is not clear what the differences between these grades are or how subjects were able to delineate this difference. As an example, when comparing definitely easy and quite easy, it is hard to say which denotes the "easier" of the two options.

Discussion- It is mentioned that the usage rate for IBD doc is higher than traditional fecal calprotectin based on your groups prior work. However, it is hard to draw this conclusion as this is accounting only for the 30 subjects who agreed to participate in the study. It doesn’t account the people that likely refused to take part in this study and perhaps wouldn’t want to perform an at home fecal calprotectin test if prescribed.

Author Response

Brief Summary: This manuscript by D’Amico et al. provides a narrative review of e-health and fecal calprotectin home testing kits. In addition, they provide their own survey study results relating to the use of home fecal calprotectin kits. This review is timely due to the impact that COVID-19 has had on healthcare and the management of IBD, as more care is shifting remotely. General Comments: The strengths of this manuscript relate to its combination of a nice review of the literature, specifically as it relates to home fecal calprotectin testing, with the additional experience of a large IBD center in this area of study. The primary limitation however is the small sample size of the survey which limits its generalizability and underpowers statistical analysis of the data. Furthermore, some improvement in the grammar and sentence structure is needed. Consideration may be given to try to increase the number of individuals who answer the survey either within your institution or by incorporating other institutions as this would strengthen the survey results.

Reply: We gratefully thank the reviewer for her/his comment. As underlined in the discussion of the revised article, we are aware that the small sample size is a limitation of our study and our data should be confirmed in a larger population also including patients from other centers. Enrolling further patients to increase the sample size would take several weeks and that is not feasible to meet the deadline of this issue.

Specific comments:

Line 34- "adhesion" should be changed to "adherence"

Reply: Change made

Line 73- "E-Health Literature Evidence" should be rephrased.

Reply: We have made the appropriate changes: “Literature evidence on the efficacy of patient care based on e-health”

Line 81- Are the "routine" outpatient visits that are referred to referring to all outpatient visits inclusive of telemedicine visits, or is this referring to additional visits? It would be helpful to clarify this point.

Reply: We thank the reviewer for the comment. We have clarified this point as follows: “In addition, the number of routine outpatient clinic and additional visits due to UC relapses was significantly lower in the telemedicine group compared to controls (35 vs 92 and 21 vs 107 respectively, p <0.0001) resulting in an overall cost saving of 189 €/patient /year.”

Line 106- "Fecal Calprotectin Home Test System" should be rephrased.

Reply: We have made the appropriate changes: “A system for monitoring fecal calprotectin at home”.

Line 125- "Fecal Calprotectin Home Test Literature Evidence" should be rephrased.

Reply: We have made the appropriate changes: “Literature evidence on fecal calprotectin home tests”

Line 170- It is interesting that in following over 2500 patients a year there are only an average of 5 calls and 10 emails per day. This seems low. Is this in addition to communication between patients and their doctors directly. If so, would add clarification about this.

Reply: We thank the reviewer for this relevant comment. We have clarified that in some cases, after an initial contact mediated by the IBD nurse, direct communication between patient and physician is also possible.

Line 195- Could more information be given on the fecal calprotectin thresholds chosen. These thresholds don’t seem to align with growing evidence for cutoffs in CD and UC.

Reply: We thank the reviewer for this comment. As specified in our "methods" section, we have used the thresholds recommended by the manufacturer. We have added the related reference: “https://www.ibdoc.net/wp-content/uploads/2014/07/IBDoc_Calprotectin-Home-Test_CF063ML-05E.pdf”.

Line 209- It is stated that 30 subjects agreed to participate. How many people were asked to participate? Should include the response rate.

Reply: This is a pilot study aimed at assessing adherence to measurement, patient satisfaction, and usability of the home test. Patients with a compatible smartphone who needed fecal calprotectin monitoring were consecutively invited to participate in the study. We have specified that all invited patients agreed to participate in the study.

Line 220- Smart phone use was broken down into completely regular, regular or quite regular use. It is unclear what the difference between these groups are. Clarification of what these designations mean would be helpful.

Reply: We thank the reviewer for this comment. In the "methods" section, we have clarified that we used a Likert scale from 1 to 7 to evaluate patient usability and satisfaction. For this reason, values ​​between 1 to 3 indicated a negative result, those between 5 and 7 a positive result, and values ​​equal to 4 represented an intermediate value. We have modified our “results” section according to this scale to simplify reading and facilitate understanding.

Line 223- It is unclear what difficulty in handling the stool means. Clarification in the manuscript regarding this would be helpful.

Reply: We have made the appropriate changes, specifying that the meaning of "handling the stool" was "sampling the stool".

Line 231- It is unclear what the difference between definitely adequate or adequate information is. Was this defined in the survey for the patients. Would recommend clarifying this in the manuscript. Similar statements are made relating to ease of use and difficulty with stool sampling as well as other survey results. Clarification of this grading scale is needed as it is not clear what the differences between these grades are or how subjects were able to delineate this difference. As an example, when comparing definitely easy and quite easy, it is hard to say which denotes the "easier" of the two options.

Reply: We thank the reviewer for this suggestion. As specified above, we have clarified the interpretation of the results according to the Likert scale model.

Discussion- It is mentioned that the usage rate for IBD doc is higher than traditional fecal calprotectin based on your groups prior work. However, it is hard to draw this conclusion as this is accounting only for the 30 subjects who agreed to participate in the study. It doesn’t account the people that likely refused to take part in this study and perhaps wouldn’t want to perform an at home fecal calprotectin test if prescribed.

Reply: We thank the reviewer for this comment. As previously mentioned, we specified that all 30 invited patients agreed to participate. Two thirds of the patients performed the measurement, while the remaining 10 patients were not compliant for several reasons: difficulty in performing the measurement alone (3/10, 30%), absence of an economic reimbursement (2, 20%), disease worsening (2, 20%), forgetfulness (2, 20%), and address change (1, 10%). Despite the small number of patients included in our study, we believe this finding may be indicative of patient adherence. Obviously these data need further confirmation in larger studies as we clarified in our discussion.

Reviewer 2 Report

The manuscript by D’Amico et al. is a combination of 1) a review of the literature related to virtual IBD clinics and fecal calprotectin (FC) home tests and 2) presentation of results from a study on the experience from Nancy University Hospital from home monitoring through FC tests.

The review seems rather incomplete and the research data from the Nancy experience is based on only 20 patients.

I would suggest to rewrite this into a review, preferably a systematic one including more thorough information on virtual IBD clinics and home FC and eventually include just a short summary of the data from the Nancy experience in the review.

Specific points

Title:

The title is somewhat misleading.  Now, the manuscript is a combination of a review and presentation of the Nancy experience.

Abstract:

It is stated that: “This pilot experience shows that a virtual calprotectin clinic doubles adhesion rate to FC and allows early intervention in IBD patients”. I cannot find strong data supporting these two statements.  In the discussion a comparison with historical controls is mentioned.

Page 3, line 67: The adherence was in one study 35%. What about other studies/populations/countries?

Generally, page 4- 8. The studies referred to are incompletely described. Included number of patients, time periods, diagnosis (UC/CD), etc. is missing.  The evaluation of the included studies should be more critical. As described now, it is mostly data speaking in favour of virtual clinics and home FC that are highlighted.

Page 4, line 78: “most patients (88%)” Did all patients in the study try both approaches?

Page 4, line 83-89, How many patients in the study? Randomised 1:1:1?

Page 5, line 92. An “a” to much.

Page 5, line 91-99. How many patients in the study?

Page 5, line 95. What is “1.26 vs 1.98”? Number of visits per …?

Page 5, line 99. The CI is higher than the value!?

Page 5, line 99-105. How many patients in the study?

Page 7, line 133. Omit “more”?

Page 7, line 138-  : How many patients? Studied for how long?

Methods:

Was the study approved by an ethical review board?

Is the study registered in a clinical trial database (i.e. clinicaltrials.gov)?

Did the participants sign informed consents?

Has the questionnaire been validated?

How often did the patients measure FC during the study period? Did all patients do it with the same interval?

Results:

The number of patients included (20) is low considering the large number of patients handle at the clinic (>2500 patients).

Page 11, line 220: hard to know what the difference is between “completely regular”, “regular” and “quite regular”.

Page 11, line 225-226: Informed when in relation to the study?

Page 11, line 241-243: I do not understand this part. Please rephrase.

Page 248, line. Suggest to rephrase “”would have liked to”

ript is rewritten into a review,

Author Response

The manuscript by D’Amico et al. is a combination of 1) a review of the literature related to virtual IBD clinics and fecal calprotectin (FC) home tests and 2) presentation of results from a study on the experience from Nancy University Hospital from home monitoring through FC tests. The review seems rather incomplete and the research data from the Nancy experience is based on only 20 patients. I would suggest to rewrite this into a review, preferably a systematic one including more thorough information on virtual IBD clinics and home FC and eventually include just a short summary of the data from the Nancy experience in the review.

Reply: We thank the reviewer for this comment but his/her suggestions are beyond the scope of our work. This is an invited review for a special issue and we provided a literature overview as requested by the editors.

Specific points

Title: The title is somewhat misleading. Now, the manuscript is a combination of a review and presentation of the Nancy experience.

Reply: We thank the reviewer for the comment. We have modified the title as follows: “Setting up a virtual calprotectin clinic in inflammatory bowel diseases: literature review and Nancy experience”

Abstract: It is stated that: “This pilot experience shows that a virtual calprotectin clinic doubles adhesion rate to FC and allows early intervention in IBD patients”. I cannot find strong data supporting these two statements. In the discussion a comparison with historical controls is mentioned.

Reply: Valid suggestion. We have modified our abstract as we have not shown that home tests allow early intervention in IBD patients. However, as underlined in our discussion, a double adherence rate was found compared to our previous work, which was the first that evaluated compliance with the traditional ELISA test in IBD patients.

Page 3, line 67: The adherence was in one study 35%. What about other studies/populations/countries?

Reply: We thank the reviewer for this comment. The study by Maréchal et al. in 2017 was the first specifically designed to assess compliance with traditional FC monitoring in IBD patients. Unfortunately, we could not find other studies reporting this relevant finding in adult IBD population.

Generally, page 4- 8. The studies referred to are incompletely described. Included number of patients, time periods, diagnosis (UC/CD), etc. is missing. The evaluation of the included studies should be more critical. As described now, it is mostly data speaking in favour of virtual clinics and home FC that are highlighted.

Reply: We thank the reviewer for this comment. We have provided all available information as requested (number of patients, time periods, diagnosis (UC/CD)).

Page 4, line 78: “most patients (88%)” Did all patients in the study try both approaches?

Reply: We thank the reviewer for the comment. We clarified that patients in the telemedicine group preferred the new approach compared with the traditional one.

Page 4, line 83-89, How many patients in the study? Randomised 1:1:1?

Reply: Thanks for this comment. We have added the number of included patients and we have specified that patients were randomly assigned 1:1:1 in the three study arms.

Page 5, line 92. An “a” to much.

Reply: We have made the appropriate change.

Page 5, line 91-99. How many patients in the study?

Reply: We have included the number of patients as requested.

Page 5, line 95. What is “1.26 vs 1.98”? Number of visits per ...?

Reply: Thank you for this comment. We have clarified that these are the number of outpatients visits per patient during a follow-up period of 12 months.

Page 5, line 99. The CI is higher than the value!?

Reply: We thank the reviewer for the comment. We have made the appropriate change.

Page 5, line 99-105. How many patients in the study?

Reply: We have included the number of patients as requested.

Page 7, line 133. Omit “more”?

Reply: Change made

Page 7, line 138- : How many patients? Studied for how long?

Reply: Thank you for this comment. We have added in the text the requested information.

Methods:

Was the study approved by an ethical review board?

Reply: Thank you for this comment. We have added data regarding the ethic committee approval: “The ethical approval code of our study was reported to the Commission Nationale de l’Informatique et des Libertés (number 1404720).”

Is the study registered in a clinical trial database (i.e. clinicaltrials.gov)?

Reply: We thank the reviewer for this comment. No, this study was not registered in a clinical trial database.

Did the participants sign informed consents?

Reply: All patients provided verbal consent for participation in the study.

Has the questionnaire been validated?

Reply: We thank the reviewer for the comment. Unfortunately our questionnaire was not validated. In the discussion, we have added the non-use of a validated questionnaire as a limitation of our study.

How often did the patients measure FC during the study period? Did all patients do it with the same interval?

Reply: Thank you for the comment. All patients were required to perform only one home test for FC measurement.

Results:

The number of patients included (20) is low considering the large number of patients handle at the clinic (>2500 patients).

Reply: Valid suggestion. We have clarified that this was a pilot study and 30 consecutive patients were enrolled.

Page 11, line 220: hard to know what the difference is between “completely regular”, “regular” and “quite regular”.

Reply: Thank you for the comment. We have reported the evaluation criteria according to the Likert scale in the methods and have modified the sentence to simplify its understanding.

Page 11, line 225-226: Informed when in relation to the study?

Reply: We have specified that all patients were informed of the existence of the FC home tes at the time of the inclusion visit.

Page 11, line 241-243: I do not understand this part. Please rephrase.

Reply: We have made the appropriate changes.

Page 248, line. Suggest to rephrase “”would have liked to”  ript is rewritten into a review

Reply: We have rephased the sentence as requested.

Reviewer 3 Report

D’Amico et. al have produced a brief review of literature on telehealth in IBD care anddata on a small pilot study at a single medical center that implemented theIBDoc home-based FC testing. This study needs major revisions in writing and has limitations due to small sample size and lack of validated assessment tools. This study also lacks critical components of clinical disease assessment, assessment of the reproducibility of home-based FC testing and correlation with lab -based FC testing.

Specific points:

--Are there studies that evaluated diagnostic accuracy of IBDoc?  
--Is there validation for the cutoff values of <100 for the IBDoc home based test for normal values?
--Was intra-individual reproducibility assessed with the home-based FC testing?
--It would be useful to correlate home-based FC testing with clinical indices (HBI, partial Mayo score), traditional lab-based ELISA and gold standard of endoscopic evaluation
--Why would 2/20 not have a prescription (?order) for the FC test?
--The lack of a validated assessment, such as the system usability scale, is a weakness. 
--The manuscript states "All patients with a compatible smartphone who agreed to monitor FC at home through IBDoc® were eligible for inclusion." How were eligible patients initially identified and contacted?  How many potential participants were contacted?  Was there any difference between those who agreed to participate and those who did not?   If recruitment took place between Sep 2018 and Dec 2019 at an IBD center of 2500 patients, it is surprising that only 30 patients agreed to participate. This raises the concern for bias.
--The study is very small, with only 20 patients participating in stool submission.

--Do the investigators know why 10 of the initial 30 participants did not submit their stool after initially agreeing to participate? 

--All patients in this study are less than 44 years of age. Were older patients offered to participate?

--It is difficult to understand the difference among the various answer choices (ie. "easy", "quite easy" and "definitely easy"). 

--Table 1:

---I recommend adding a column to Table 1 containing the patient characteristics of all IBD patients at Nancy University Hospital.  sessment (partial mayo and HBI for example)
---Include disease characteristics (extent, phenotype), IBD medication, clinical disease assessment (partial mayo and HBI for example)

--Include in discussion that more studies to compare IBDoc to traditional ELISA as well as endoscopic disease are needed before these tests are widely adapted and used in clinical practice.

Minor Points:·  

  • Abstract: Consider, “assessment” in place of the term “dosage” of fecal calprotectin at home as this is more appropriate language
  • Abstract: Change “adhesion” to “adherence”
  • Introduction: Change “considerably increasing” to “increasing considerably”
  • Introduction: Include CT under imaging techniques for IBD
  • Introduction: Change “inconvertible” to “conceivable”
  • E-Health Literature Evidence line 77: change consisted “in” to “of”
  • Results line 253-254: Rephrase, this is hard to read
  • Discussion line 257: Change “constantly” to “consistently”
  • Discussion line 265 change “a double” to “double the”

Author Response

D’Amico et. al have produced a brief review of literature on telehealth in IBD care and data on a small pilot study at a single medical center that implemented the IBDoc home-based FC testing. This study needs major revisions in writing and has limitations due to small sample size and lack of validated assessment tools. This study also lacks critical components of clinical disease assessment, assessment of the reproducibility of home-based FC testing and correlation with lab -based FC testing.

Reply: We thank the reviewer for his/her comment, but many of the aspects indicated such as the evaluation of reproducibility and the comparison with other FC tests were outside our goal. Our work is a review on the e-health strategy with a focus on FC home monitoring in which we have reported the data of our small pilot study.

Specific points:

--Are there studies that evaluated diagnostic accuracy of IBDoc? 

Reply: We thank the reviewer for this comment. We have added the diagnostic accuracy of IBDoc evaluated in the study by Bello and colleagues: “the home test had high diagnostic accuracy in predicting a FC> 300μg/g with the ELISA test (sensitivity: 89.8%, specificity: 95.5%, negative predictive value: 91.4%, and positive predictive value: 94.6%)”

--Is there validation for the cutoff values of <100 for the IBDoc home based test for normal values?

Reply: We thank the reviewer for the question. The the cutoff of <100 for the IBDoc test for normal values was validated in the study by Bello and colleagues. We have added the reference in our “methods” section.

--Was intra-individual reproducibility assessed with the home-based FC testing?

Reply: Valid suggestion. We have added reproducibility data for both IBDoc and CalproSmart as requested.

--It would be useful to correlate home-based FC testing with clinical indices (HBI, partial Mayo score), traditional lab-based ELISA and gold standard of endoscopic evaluation

Reply: We thank the reviewer for these suggestions, but our aim was not to evaluate the operating characteristics of the IBDoc nor its correlation with clinical, endoscopic, or other measurement tests. Our goal was to assess adherence, satisfaction, and usability of the home test.

--Why would 2/20 not have a prescription (?order) for the FC test?

Reply: We thank the reviewer for the question. Two patients had lost their prescription. We did not include this data in the text, as it seemed to us of little relevance for the purpose of the study.

--The lack of a validated assessment, such as the system usability scale, is a weakness.

Reply: We thank the reviewer for this comment. We have added this point as a limitation of our study.

--The manuscript states "All patients with a compatible smartphone who agreed to monitor FC at home through IBDoc® were eligible for inclusion." How were eligible patients initially identified and contacted?  How many potential participants were contacted?  Was there any difference between those who agreed to participate and those who did not?   If recruitment took place between Sep 2018 and Dec 2019 at an IBD center of 2500 patients, it is surprising that only 30 patients agreed to participate. This raises the concern for bias.

Reply: Thank you for this comment. We have specified that 30 consecutive patients were invited to participate in the study during routine outpatient clinic. Patients with incompatible smartphones and a previous prescription for FC measurement with a standard technique were excluded. All invited patients accepted to participate in the study. In addition, enrollments were performed only one day a week, when the dedicated nurse was available to provide patients with the home test information.

--The study is very small, with only 20 patients participating in stool submission.

Reply: Thank you for this comment. We are aware of this limitation and have clearly stated it in our discussion.

--Do the investigators know why 10 of the initial 30 participants did not submit their stool after initially agreeing to participate?

Reply: We thank the reviewer for this relevant comment. We have addded the causes of non-adherence as requested: difficulty in performing the measurement alone (3/10, 30%), absence of an economic reimbursement (2, 20%), disease worsening (2, 20%), forgetfulness (2, 20%), and address change (1, 10%).

--All patients in this study are less than 44 years of age. Were older patients offered to participate?

Reply: No age limit was imposed. Elderly patients were eligible. We enrolled the first 30 patients with compatible smartphones, without a previous prescription for the FC dosage.

--It is difficult to understand the difference among the various answer choices (ie. "easy", "quite easy" and "definitely easy").

Reply: We thank the reviewer for this comment. As indicated in the response to reviewer 1's comment, we have modified the description of the results to make the interpretation of the results more understandable and easier.

--Table 1:

---I recommend adding a column to Table 1 containing the patient characteristics of all IBD patients at Nancy University Hospital.  assessment (partial mayo and HBI for example)

Reply: We thank the reviewer for this suggestion. We have added data on clinical assessment of disease in Table 1 as recommended.

---Include disease characteristics (extent, phenotype), IBD medication, clinical disease assessment (partial mayo and HBI for example)

Reply: We thank the reviewer for the comment. We have included disease characteristics in Table 1 as requested.

--Include in discussion that more studies to compare IBDoc to traditional ELISA as well as endoscopic disease are needed before these tests are widely adapted and used in clinical practice.

Reply: We have appreciated the reviewer’s comment and we have added the following sentence in our discussion: “Additional studies comparing home tests and traditional ELISA test sas well as the correlation with endoscopic disease activity are needed before these tests are widely used in clinical practice”.

Minor Points:· 

Abstract: Consider, “assessment” in place of the term “dosage” of fecal calprotectin at home as this is more appropriate language

Reply: Change made

Abstract: Change “adhesion” to “adherence”

Reply: Change made

Introduction: Change “considerably increasing” to “increasing considerably”

Reply: Change made

Introduction: Include CT under imaging techniques for IBD

Reply: We have included computed tomography as an available imaging technique for IBD as recommended.

Introduction: Change “inconvertible” to “conceivable”

Reply: Change made

E-Health Literature Evidence line 77: change consisted “in” to “of”

Reply: Change made

Results line 253-254: Rephrase, this is hard to read

Reply: Change made

Discussion line 257: Change “constantly” to “consistently”

Reply: Change made

Discussion line 265 change “a double” to “double the”

Reply: Change made

Reviewer 4 Report

医療関係者にとって、感染防止対策として非常に役立つ制度だと思います。残念ながら、いくつかのケースがあります。今後も蓄積してください。また、それが疾病活動の抑制に影響を及ぼしているかどうかも調査していただきたい。

Author Response

I think this is a very useful system for medical personnel to prevent infection. Unfortunately, there are some cases. Please continue to accumulate. Also, please investigate whether it affects the control of disease activity.

Reply: We thank the reviewer for her/his comment and interest in this topic. However, this is an invited review and our aim was to provide a literature review including data of our pilot study. As specified in the response to reviewer 1, enrolling further patients to increase the sample size would take several weeks and that is not feasible to meet the deadline of this issue. In addition, the aim of our pilot study was to assess test adherence, usability, and patient satisfaction with the IBDoc. Assessing diagnostic accuracy and the impact of testing on disease control is beyond the scope of our work.

Round 2

Reviewer 1 Report

        1. Introduction- would remove the term "literature evidence" and make it just "literature"

        2. Title "LITERATURE EVIDENCE ON THE EFFICACY OF PATIENT CARE BASED ON E- HEALTH" needs to be changed grammatically.

3. The title "LITERATURE EVIDENCE ON FECAL CALPROTECTIN HOME TESTS" needs to be changed grammatically as well- ie. "Evidence for fecal calprotectin home testing" or just "Fecal Calprotectin Home Testing"

4. Line 129 should read "non-compliant patients".

5. Line 146 should read "to use the extraction tool"

6. Line 196 the reference citation is bracketed. These brackets should be removed.

7. Line 205 should read "candidate for a"

8. Line 212 should read "3/10" to keep this consistent with the rest of the sentence structure.

9. Line 243, there is no delineation of satisfaction in the first part of the sentence.

10. Were the survey questions presented in a way that subjects could differentiate and state their degree of agreement in the answer choices? As an example, it is still not clear which is easier, -definitely easy, easy, or quite easy- which would be the three choices presented to a subject. While I understand a likert scale was used with numbers assigned for the analysis of the scale, I’m not sure a reader could differentiate which degree of easy is in fact easier. This extends to the other questions delineating degrees of agreement or disagreement as outlined in table 2.

11. Very minor improvements in grammar are needed to improve readability.

Author Response

  1. Introduction- would remove the term "literature evidence" and make it just "literature"

Reply: Thank you for your suggestion. We have made the appropriate change.

  1. Title "LITERATURE EVIDENCE ON THE EFFICACY OF PATIENT CARE BASED ON E- HEALTH" needs to be changed grammatically.

Reply: We thank the reviewer for this comment. We have modified the title as follows: “EFFICACY OF PATIENT CARE BASED ON THE E-HEALTH”.

  1. The title "LITERATURE EVIDENCE ON FECAL CALPROTECTIN HOME TESTS" needs to be changed grammatically as well- ie. "Evidence for fecal calprotectin home testing" or just "Fecal Calprotectin Home Testing"

Reply: Thank you for this comment. We have modified the title as recommended.

  1. Line 129 should read "non-compliant patients".

Reply: Change made

  1. Line 146 should read "to use the extraction tool"

Reply: Change made

  1. Line 196 the reference citation is bracketed. These brackets should be removed.

Reply: Change made

  1. Line 205 should read "candidate for a"

Reply: Change made

  1. Line 212 should read "3/10" to keep this consistent with the rest of the sentence structure.

Reply: We have modified the sentence, using the same structure throughout the manuscript as recommended.

  1. Line 243, there is no delineation of satisfaction in the first part of the sentence.

Reply: Thank you for this comment. We have appropriately modified the sentence.

  1. Were the survey questions presented in a way that subjects could differentiate and state their degree of agreement in the answer choices? As an example, it is still not clear which is easier, -definitely easy, easy, or quite easy- which would be the three choices presented to a subject. While I understand a likert scale was used with numbers assigned for the analysis of the scale, I’m not sure a reader could differentiate which degree of easy is in fact easier. This extends to the other questions delineating degrees of agreement or disagreement as outlined in table 2.

Reply: Thank you for this comment. As specified in the "methods" section, patients were asked to respond with a value between 1 and 7. It was then specified that the value of 1 corresponded to the lowest value, while the value of 7 corresponded to the highest value. A brief description was associated with the numerical value. The presence of the number and the description helped the patient to understand the answers. To help the readers, we have added the numbers next to the description in the tables to avoid any interpretation errors.

  1. Very minor improvements in grammar are needed to improve readability.

Reply: We thank the reviewer for the comment. We have reviewed and appropriately corrected the minor errors in the manuscript as recommended.